# FASTColor – Full-color Amplitude Surrogate Toolkit for QCD

Javier Mariño Villadamigo[1], Rikkert Frederix[2],
Tilman Plehn[1,3], Timea Vitos[4,5], and Ramon Winterhalder[6]

**1** Institut für Theoretische Physik, Universität Heidelberg, Germany
**2** Department of Physics, Lund University, Sweden
**3** Interdisciplinary Center for Scientific Computing (IWR), Universität Heidelberg, Germany
**4** Department of Physics and Astronomy, Uppsala University, Sweden
**5** Institute for Theoretical Physics, ELTE University Budapest, Hungary
**6** TIFLab, Universitá degli Studi di Milano & INFN Sezione di Milano, Italy

September 10, 2025

## Abstract

**High-multiplicity events remain a bottleneck for LHC simulations due to their computational cost. We present a ML-surrogate approach to accelerate matrix element reweighting from leading-color (LC) to full-color (FC) accuracy, building on recent advancements in LC event generation. Comparing a variety of modern network architectures for representative QCD processes, we achieve speed-up of around a factor two over the current LC-to-FC baseline. We also show how transformers learn and exploit underlying symmetries, to improve generalization. Given the gained trust in trained networks and developments in learned uncertainties, the LC-to-FC approach will eventually benefit further from not needing a final classic unweighting step.**

# 1  Introduction

The success of the LHC physics program crucially depends on precise theoretical predictions via first-principle simulations. With the upcoming High-Luminosity LHC (HL-LHC), the demand for accurate simulations across vast regions of phase space with large multiplicities will increase dramatically. Multi-jet final states play a central role in both Standard Model measurements and searches for new physics, but their theoretical description is computationally challenging. At large multiplicities, the evaluation of QCD matrix elements becomes expensive, as the number of contributing Feynman diagrams and color-correlated amplitudes grows, usually, factorially with the number of external partons. Even though color-summed squared matrix elements can be computed with only exponential complexity with the number of external partons [1], numerical Monte Carlo integration becomes progressively less efficient, and unweighting efficiencies deteriorate with increasing multiplicity [2–7]. This poses a severe challenge for event generators, which in turn are indispensable for connecting quantum field theory to LHC data.

A range of classical techniques has been developed to address these challenges. Phase-space integration efficiency is improved by adaptive importance sampling algorithms such as VEGAS [8–10] and PARNI [11], and by multi-channeling methods [12,13], often combined with optimized phase-space parametrizations that align integration variables with the propagator structure of the contributing Feynman diagrams [14–18]. Beyond integration, the factorial growth of amplitudes can be tamed through recursive relations and Monte Carlo sampling of helicities and colors [19–27]. Despite these advances, the scaling of fixed-order QCD predictions at high multiplicity remains prohibitive, motivating further innovations.

A promising strategy is to restrict matrix elements to their leading-color (LC) contributions, which capture the dominant terms in the $1/N_c$ expansion at a fraction of the computational cost. While LC predictions often provide a good proxy for full-color (FC) results, they are not sufficient for precision studies. This motivates a two-step strategy [28]: events are first generated at LC accuracy and then reweighted to the exact FC matrix elements. This LC-to-FC unweighting approach yields substantial gains over direct FC generation, with per-mille level efficiency even for $2 \to 7$ processes. It makes very high multiplicity channels computationally tractable and opens the door to further improvements.

A way to further improve this strategy is modern machine learning (ML), a transformative methodology all across particle physics [29, 30]. ML methods are being applied in phase-space sampling [31–42], approximation of complex scattering amplitudes [43–54], generation of complete collider events [55–61], and detector simulation with unprecedented speed [62–81]*. In many applications, ML surrogates provide a natural way to embed learned approximations inside existing Monte Carlo pipelines [83]. For LC-to-FC event generation, surrogate models can be trained to predict the reweighting ratio, thereby reducing the reliance on repeated evaluations of expensive FC amplitudes and offering significant computational savings. Similar surrogate–reweighting strategies have already been explored in related contexts [84–86]. Building on these ideas, we demonstrate how ML surrogates can further accelerate the LC-to-FC pipeline while retaining FC accuracy.

The paper is organized as follows: In Sec. 2, we review the leading-color approximation and introduce our machine-learned three-step unweighting procedure. In Sec. 3, we discuss how underlying symmetries of the dataset can affect the learning behavior of the network. Afterwards, in Sec. 4, we quantify the overall performance in terms of an effective gain factor for representative classes of QCD processes. We conclude with an outlook in Sec. 5.

---

*We refer to the Living Review of ML in particle physics [82] for up-to-date developments at the intersection of these two fields.

## 2  Machine-learned event generation

Rather than attempting to generate events directly with full-color (FC) accuracy, we can improve event generation by decomposing the task into successive steps of increasing accuracy. In the first step, we produce events in the leading-color (LC) approximation, which captures their dominant structure at reduced computational cost. Next, the relative FC corrections are incorporated through a machine-learning (ML) surrogate for the corresponding reweighting factor. This multi-stage strategy balances efficiency and accuracy, providing a scalable path towards high-precision event samples.

We consider processes involving only gluons, processes involving one quark line, and processes involving two quark lines. They are of the form $ab \to 12 \ldots n$ with $n \in \{4, 5, 6, 7\}$. Since, at LC we can consider each color ordering separately, we restrict ourselves to a representative subset of all possible color orderings, denoted by a string of particle numbers between brackets. For an easier overview, in the color orderings, the anti-quark labels are denoted with an overhead bar, and the quark labels are underlined. Note that the processes are symmetry crossed when considering the color ordering. The full set of processes we target is

$$
\begin{aligned}
\text{all-gluon:} \quad & gg \to ng & (ab123\ldots n) & \qquad (1)\\
\text{one-quark line:} \quad & gg \to d\bar{d} + (n-2)g & (\underline{1}ab3\ldots n\bar{2}) & \\
& d\bar{d} \to ng & (\underline{a}123\ldots n\bar{b}) & \qquad (2)\\
\text{two-quark line:} \quad & d\bar{d} \to u\bar{u} + (n-2)g & (\underline{b}\bar{a}\underline{1}3\ldots n\bar{2}) & \quad [\text{CO1}]\\
& & (\underline{b}\bar{2}\underline{1}3\ldots \bar{n}a) & \quad [\text{CO2}] \qquad (3)\\
& gg \to d\bar{d}u\bar{u} + (n-4)g & (\underline{1}ab\bar{2}\underline{3}n\ldots 5\bar{4}) & \quad [\text{CO1}]\\
& & (\underline{1}ab\bar{4}\underline{3}n\ldots 5\bar{2}) & \quad [\text{CO2}] \qquad (4)
\end{aligned}
$$

For processes involving only gluons and one quark line, we choose one representative color ordering of the amplitude, written in Eqs.(1) and (2). For processes with two quark lines, we choose two orderings CO1 and CO2, as indicated in Eq.(4).

In each of the color orderings, $640 \times 10^3$ phase-space points passing the generation cuts are used to set up the grids for importance sampling. These grids remain fixed, and we use a set of $6.4 \times 10^6$ events to determine the maximum weight for the unweighting of the LC-accurate events. This is used to compute the unweighting efficiency of the leading color matrix elements. In the LC-based unweighting approach, all unweighted LC events accepted by the hit-or-miss method are further reweighted according to their FC weight. In a second unweighting step, we obtain a set of FC events.

### 2.1  The leading-color approximation

The evaluation of QCD matrix elements is computationally expensive, and its cost grows rapidly with the number of external particles. This complexity manifests itself in a large variance of the event weights obtained during phase-space integration. Unweighted events are typically obtained via a simple hit-or-miss algorithm, in which each event is accepted with probability $w/w_{\max}$, where $w_{\max}$ is the maximum event weight in the sample. The unweighting efficiency for a set of $N$ events is defined as [34]

$$
\epsilon_{\text{uw}} = \frac{\langle w \rangle_N}{w_{\max}}, \qquad (5)
$$

Rare tail events with very large weights, $w_{\max} \gg \langle w \rangle_N$, lead to a very low unweighting efficiency and become problematic for costly matrix element evaluations. As we will discuss in

Sec. 2.2, $w_{\text{max}}$ does not have to be chosen as the absolute maximum. Instead, a percentile-based maximum weight can increase efficiency at the expense of keeping a controlled fraction of overweights.

A well-established strategy to improve costly evaluations is the LC approximation [28]. It splits event generation into first generating events from an LC approximation of the matrix element and then reweighting them to FC. It reduces the variance of the event weights and improves the efficiency of the phase-space integration. We derive this two-step procedure in detail, starting by writing the FC cross section as

$$\sigma_{\text{FC}} = \int d\Phi \sum_h \sum_{i,j} \mathcal{A}_{i,h}(x) C_{ij} \mathcal{A}_{j,h}^*(x) = \int d\Phi \sum_h |\mathcal{M}(x,h)|^2 \, , \tag{6}$$

where $\mathcal{A}_{i,h}(x)$ denotes the $i$-th color-ordered or dual amplitude for helicity $h$, and $C_{ij}$ is the color matrix. The LC approximation at fixed helicity is defined as

$$|\mathcal{M}_{\text{LC}}(x,h)|^2 = \sum_i C_{ii} |\mathcal{A}_{i,h}(x)|^2 \, . \tag{7}$$

Inserting a unit factor expressed in terms of this quantity yields

$$\sigma_{\text{FC}} = \int d\Phi \sum_h |\mathcal{M}_{\text{LC}}(x,h)|^2 \, r_{\text{LC}\to\text{FC}}(x,h) \qquad \text{with} \qquad r_{\text{LC}\to\text{FC}}(x,h) \equiv \frac{|\mathcal{M}(x,h)|^2}{|\mathcal{M}_{\text{LC}}(x,h)|^2} \, . \tag{8}$$

Expanding the FC expression (Eq. 6) into color flows $i$ gives

$$\sigma_{\text{FC}} = \sum_i \int d\Phi \sum_h C_{ii} |\mathcal{A}_{i,h}(x)|^2 \, r_{\text{LC}\to\text{FC}}(x,h) \, . \tag{9}$$

For each color flow $i$, we can define

$$|\mathcal{M}_{\text{LC}}^i(x)|^2 = \sum_h C_{ii} |\mathcal{A}_{i,h}(x)|^2 \qquad \text{and} \qquad p_i(h|x) = \frac{C_{ii} |\mathcal{A}_{i,h}(x)|^2}{|\mathcal{M}_{\text{LC}}^i(x)|^2} \, , \tag{10}$$

where $p_i(h|x)$ denotes the relative size of the LC amplitude at fixed helicity $h$. This gives

$$\sigma_{\text{FC}} = \sum_i \int d\Phi \, |\mathcal{M}_{\text{LC}}^i(x)|^2 \sum_h p_i(h|x) \, r_{\text{LC}\to\text{FC}}(x,h)$$

$$\approx \sum_i \int d\Phi \, |\mathcal{M}_{\text{LC}}^i(x)|^2 \left\langle r_{\text{LC}\to\text{FC}}(x,h) \right\rangle_{h\sim p_i(h|x)} \, . \tag{11}$$

In the last line the helicity sum is estimated by drawing only a single helicity $h$ for each phase-space point $x$. In practice, this corresponds to the following steps:

1. generate LC events for a single color flow with weight $|\mathcal{M}_{\text{LC}}^i(x)|^2$;
2. assign a helicity $h$ with probability $p_i(h|x)$;
3. and reweighting the event with $r_{\text{LC}\to\text{FC}}(x,h)$.

Although only a single helicity configuration is sampled per event, the averaging over many phase-space points $x$ reproduces the full helicity sum in Eq.(6).

It is worth expanding on why helicities are treated differently at the two stages. For the parton multiplicities considered here (up to $n \simeq 7$), explicitly summing over helicities at each phase-space point reduces the variance of the event weights at LC and improves the unweighting efficiency [28]. For the subsequent FC reweighting such a helicity sum would be prohibitively expensive, and it is more efficient to sample a single helicity configuration for each phase-space point. This strategy combines variance reduction at the LC level with computational efficiency at the FC level.

## 2.2  Multi-stage unweighting

Building on the LC approximation, we refine the two-step unweighting [28] by inserting a third step. For this, we train a simple regression network for the LC→FC reweighting factor

$$r_{\text{surr}}(x,h) \approx r_{\text{LC}\to\text{FC}}(x,h) = \frac{|\mathcal{M}(x,h)|^2}{|\mathcal{M}_{\text{LC}}(x,h)|^2}\,. \tag{12}$$

This neural network surrogate provides an efficient estimate of the FC correction, which we incorporate as an additional acceptance step in the unweighting pipeline. For completeness, we recall the baseline LC method in Alg. 1, and then present our three-step algorithm, highlighting the ML surrogate as the new step to further improve efficiency while retaining full FC accuracy.

We define the average execution time of the LC algorithm $T_{\text{LC}}$, and the effective gain factor of the surrogate unweighting with respect to the LC-based unweighting as [84]

$$f^{\text{eff}} = \frac{T_{\text{LC}}}{T_{\text{surr}}}. \tag{13}$$

We can formulate the execution time of an unweighting algorithm to take into account timing, efficiency, and statistical power. For an unweighting algorithm with $M$ steps, each with an average evaluation time $\langle t_i \rangle$ and unweighting efficiency $\epsilon_i$, the total execution time to generate $N$ events is

$$T^{(M\text{-step})} = N \sum_{i=1}^{M} \frac{\langle t_i \rangle}{\prod_{j=i}^{M} \epsilon_j}. \tag{14}$$

The total execution time for the two-step LC-based unweighting of Alg. 1 is then

$$T_{\text{LC}} = N \left[ \frac{\langle t_{\text{LC}} \rangle}{\epsilon_{\text{LC}} \epsilon_r} + \frac{\langle t_{\text{FC}} \rangle}{\epsilon_r} \right], \tag{15}$$

where $\epsilon_{\text{LC}}$ represents the unweighting efficiency of the leading color matrix elements $|\mathcal{M}_{\text{LC}}^i(x)|^2$, and $\epsilon_r$ is the efficiency of unweighting against the ratio $r_{\text{LC}\to\text{FC}}$. The average evaluation time $\langle t_{\text{LC}} \rangle$ corresponds to the evaluation of a helicity-summed but color-ordered LC matrix element

---

**Algorithm 1:** LC-based unweighting for single color flow $i$

---

**while** true **do**

    generate phase-space point $x$;

    calculate LC event weight $w_{\text{LC}}^i = |\mathcal{M}_{\text{LC}}^i(x)|^2$;

    generate uniform random number $z_1 \in [0,1)$;

    # first unweighting step

    **if** $w_{\text{LC}}^i > z_1 \cdot w_{\text{LC,max}}^i$ **then**

        sample helicity $h \sim p_i(h|x)$;

        calculate single-helicity FC event weight $|\mathcal{M}(x,h)|^2$;

        determine ratio $r(x,h) = |\mathcal{M}(x,h)|^2/|\mathcal{M}_{\text{LC}}(x,h)|^2$;

        generate uniform random number $z_2 \in [0,1)$;

        # final unweighting step

        **if** $r > z_2 \cdot r_{\text{max}}$ **then**

            **return** $x$

---

---

**Algorithm 2:** Surrogate-based unweighting for color flow $i$

**while** true **do**

    generate phase-space point $x$;

    calculate LC event weight $w_{\text{LC}}^i = |\mathcal{M}_{\text{LC}}^i(x)|^2$;

    generate uniform random number $z_1 \in [0,1)$;

    # first unweighting step

    **if** $w_{\text{LC}}^i > z_1 \cdot w_{\text{LC,max}}^i$ **then**

        sample helicity $h \sim p_i(h|x)$;

        evaluate surrogate $r_{\text{surr}}(x,h)$;

        generate uniform random number $z_2 \in [0,1)$;

        # second unweighting step

        **if** $r_{\text{surr}} > z_2 \cdot r_{\text{surr,max}}$ **then**

            calculate single-helicity FC event weight $|\mathcal{M}(x,h)|^2$;

            determine ratio $r(x,h) = |\mathcal{M}(x,h)|^2/|\mathcal{M}_{\text{LC}}(x,h)|^2$;

            determine ratio $t(x,h) = r(x,h)/r_{\text{surr}}(x,h)$;

            generate uniform random number $z_3 \in [0,1)$;

            # final unweighting step

            **if** $t > z_3 \cdot t_{\text{max}}$ **then**

                **return** $x$ and $\widetilde{w} = \max\left(1, \frac{t}{t_{\text{max}}} \max\left(1, \frac{r_{\text{surr}}}{r_{\text{surr,max}}}\right)\right)$

---

$|\mathcal{M}_{\text{LC}}^i(x)|^2$, while $\langle t_{\text{FC}} \rangle$ corresponds to the single-helicity but color-summed FC matrix element $|\mathcal{M}(x,h)|^2$.

For the reweighting ratio $r_{\text{LC}\to\text{FC}}(x,h)$ we also need to evaluate the single-helicity but color-summed LC matrix elements $|\mathcal{M}_{\text{LC}}(x,h)|^2$. In practice, they are given by the diagonal entries of $|\mathcal{M}(x,h)|^2$, so their evaluation comes at no additional cost.

Our three-step surrogate approach in defined in Alg. 2. After the first step, we introduce an additional unweighting against the surrogate trained to regress $r_{\text{surr}}$. When we evaluate this surrogate, the LC matrix element has already been evaluated, the helicity of the phase-space point is defined, so we can evaluate the surrogate for a given helicity. In the presence of overweights $\widetilde{w} > 1$, this unweighting produces a sample with reduced statistical power compared to the LC-based method. For an equally powerful unweighted sample, we generate more events, as quantified by the Kish effective sample size [87]

$$N^{\text{eff}} = \frac{(\sum_i \widetilde{w})^2}{\sum_i \widetilde{w}^2} \equiv \alpha N \,. \tag{16}$$

It corresponds to a fraction $\alpha$ of the nominal number of events $N$, with $\alpha = 1$ in the limit of zero overweights. This means that with overweights we have to generate $1/\alpha$ times more events for an unweighted sample with equal statistical power.

The total execution time for the three-step ML unweighting can be written as

$$T_{\text{surr}} = \frac{N}{\alpha} \left[ \frac{\langle t_{\text{LC}} \rangle}{\epsilon_{\text{LC}} \epsilon_{\text{surr}} \epsilon_t} + \frac{\langle t_{\text{surr}} \rangle}{\epsilon_{\text{surr}} \epsilon_t} + \frac{\langle t_{\text{FC}} \rangle}{\epsilon_t} \right] \,, \tag{17}$$

where $\epsilon_{\text{surr}}$ and $\epsilon_t$ are the unweighting efficiencies against $r_{\text{surr}}$ and $t = r/r_{\text{surr}}$, respectively,

and $\langle t_{\text{surr}} \rangle$ is the average evaluation time of the surrogate. The effective gain factor reads

$$f^{\text{eff}} = \alpha \, \frac{\dfrac{\epsilon_t}{\epsilon_{\text{LC}}} \langle t_{\text{LC}} \rangle + \epsilon_t \langle t_{\text{FC}} \rangle}{\dfrac{\epsilon_r}{\epsilon_{\text{surr}} \epsilon_{\text{LC}}} \langle t_{\text{LC}} \rangle + \dfrac{\epsilon_r}{\epsilon_{\text{surr}}} \langle t_{\text{surr}} \rangle + \epsilon_r \langle t_{\text{FC}} \rangle} \; . \tag{18}$$

**Choosing percentile-based maxima**

As mentioned before, we do not need to employ a (numerically pre-determined) strict maximum of the sample at each unweighting step. Instead, we can choose a lower value and accept a number of events with an overweight $\widetilde{w}_i > 1$. This is acceptable as long as the final set of weights fulfills basic criteria. This is typically controlled by imposing a minimum of acceptable $\alpha$ values. It is particularly useful when the variable against which the unweighting is performed has a very small variance, because reducing the maximum allows us to increase the efficiency significantly, without overweighting many events.

This is the typical situation for precision surrogates. Rather than strict sample maxima, we set stage-wise thresholds $r_{\text{surr,max}}$ and $t_{\text{max}}$ by percentiles to boost the unweighting efficiencies while controlling $\alpha$. Identifying the set of $N$ weights $\{w_i\}_{i=1}^N$ with $r_{\text{surr}}$ or $t$ weights, we sort them $w_1 \geq \cdots \geq w_N$, and define the $p$-percentile maximum as

$$w_{\text{max}}^p \equiv \min \left( w_j \, \middle| \, \sum_{i=j+1}^N w_i < (1-p) \sum_{i=1}^N w_i \right). \tag{19}$$

We scan $p \in \{0.4, 0.425, \ldots, 0.975, 1.00\}$ independently for $r_{\text{surr}}$ and $t$, compute the resulting efficiencies and the effective gain factor $f^{\text{eff}}$, and select the configuration that *maximizes* $f^{\text{eff}}$ while keeping $\alpha \geq 0.995$. The results for the gain factors obtained are detailed in Sec. 4.

## 2.3 Symmetry-aware architectures

The ratio between FC and LC matrix elements should be easy to learn, as the network does not need to encode the full complex physics structure. For instance for the $gg \to 4g$ process, we find that the accuracy in regressing $r_{\text{LC} \to \text{FC}}$ is typically three orders of magnitude larger than in regressing the FC amplitude.

Many network architectures have been employed for amplitude regression, MLPs [51], transformers [50], and graph networks [53]. Lorentz-equivariant architectures, such as L-GATr [50] LLoCa [53] haven significantly boosted the performance. We compare four architectures, each with their own strengths and weaknesses.

**MLP with Lorentz-invariant features** We use a fully connected feed-forward network that takes as input the logarithm of the four-momenta scalar products

$$z_{ij} = \log x_i x_j \,, \tag{20}$$

between the particle pairs. The global helicity configuration of an event is fed to the network via a look-up table coupled to a learned embedding with an embedding dimension of 64, i.e. $f(k_h) \in \mathbb{R}^{64}$, where $k_h \in \{0, \ldots, N_h - 1\}$ indexes the $N_h$ distinct global helicity configurations present in a given dataset.

**GNN with Lorentz-invariant edges** Inspired by the non-equivariant GNN in Ref. [53] we construct a directed complete graph on the $N$ particles. The edge connecting particles $i$ and $j$ is the scalar product $e_{ij} = x_i x_j$, similar to the MLP. The initial node state $n_i^{(0)}$ combines the helicity $h$ of particle $i$, its four-momentum $x$ and an appended one-hot vector encoding node index,

$$n_i^{(0)} = [x_i, h_i, o_i] \in \mathbb{R}^{5+N} \qquad o_i = \text{one\_hot}(i) \in \mathbb{R}^N \qquad i = 1, \dots, N . \tag{21}$$

The message passing is updated via MLPs $\phi_e^{(\ell)}$ acting on edge features

$$\phi_e^{(\ell)} : \mathbb{R}^{2(5+N)+1} \to \mathbb{R}^d \qquad m_{i \leftarrow j}^{(\ell)} = \phi_e^{(\ell)}\left([n_i^{(\ell)}, n_j^{(\ell)}, e_{ij}]\right) \qquad \ell = 0, \dots, L-1 , \tag{22}$$

with $L = 4$ the number of MLP layers, and $d = 128$ the hidden dimension. The node state is updated via MLPs $\phi_n^{(\ell)}$ acting on node features, as a function on its previous state and the aggregated messages,

$$\phi_n^{(\ell)} : \mathbb{R}^{(5+N)+d} \to \mathbb{R}^{5+N} \qquad n_i^{(\ell+1)} = \phi_n^{(\ell)}\left([n_i^{(\ell)}, a_i^{(\ell)}]\right) \qquad a_i^{(\ell)} = \sum_{j \neq i} m_{i \leftarrow j}^{(\ell)} . \tag{23}$$

Finally, after $L$ layers, we average over initial and final state particles separately, and use a final MLP with 8 layers and hidden dimension of 128 to obtain the final prediction for the reweighting factor

$$n_I = \frac{1}{2} \sum_{i=1}^{2} n_i^{(L)} \qquad n_F = \frac{1}{N-2} \sum_{i=3}^{N} n_i^{(L)} \qquad r_{\text{surr}}(x, h) = \Phi([n_I, n_F]) . \tag{24}$$

**Transformer** We feed the particle four-momenta and helicity to the standard PYTORCH [88] implementation. We form position-encoded tokens containing the per-particle features, similarly to the GNN. The encoder is built with a network dimension of 48, 4 heads, and a feed-forward neural network of 7 layers with hidden dimension of 170. We stack a final MLP with 3 layers to output the prediction for $r_{\text{surr}}$.

**L-GATr** The Lorentz-equivariant Geometric Algebra Transformer is described in Ref. [50]. For L-GATr, we append the vector of helicities to the global tokens for each event,

$$\vec{h} = [h_1, \dots, h_N] . \tag{25}$$

**Overall setup** Except for the MLP, which uses Lorentz invariants, all networks use four-momenta divided by the global standard deviation as inputs. The regression target $r_{\text{LC} \to \text{FC}}$ is first transformed with the hyperbolic arctangent, and then normalized. We use a standard Mean-Squared-Error (MSE) loss function to optimize the networks. All networks are trained on 600k events, with 100k for validation and 300k for testing; results are reported on the test set. Each training uses early stopping (patience 60), the Adam [89] optimizer with an initial learning rate of $10^{-3}$, combined with a learning-rate schedule that dynamically updates the learning rate during training. All networks have roughly 200k parameters.

## 3  Symmetry-learning dynamics

In a brief intermezzo, we demonstrate that the transformer indeed learns symmetries, which have become a guiding principle for network performance and interpretability especially in particle physics.

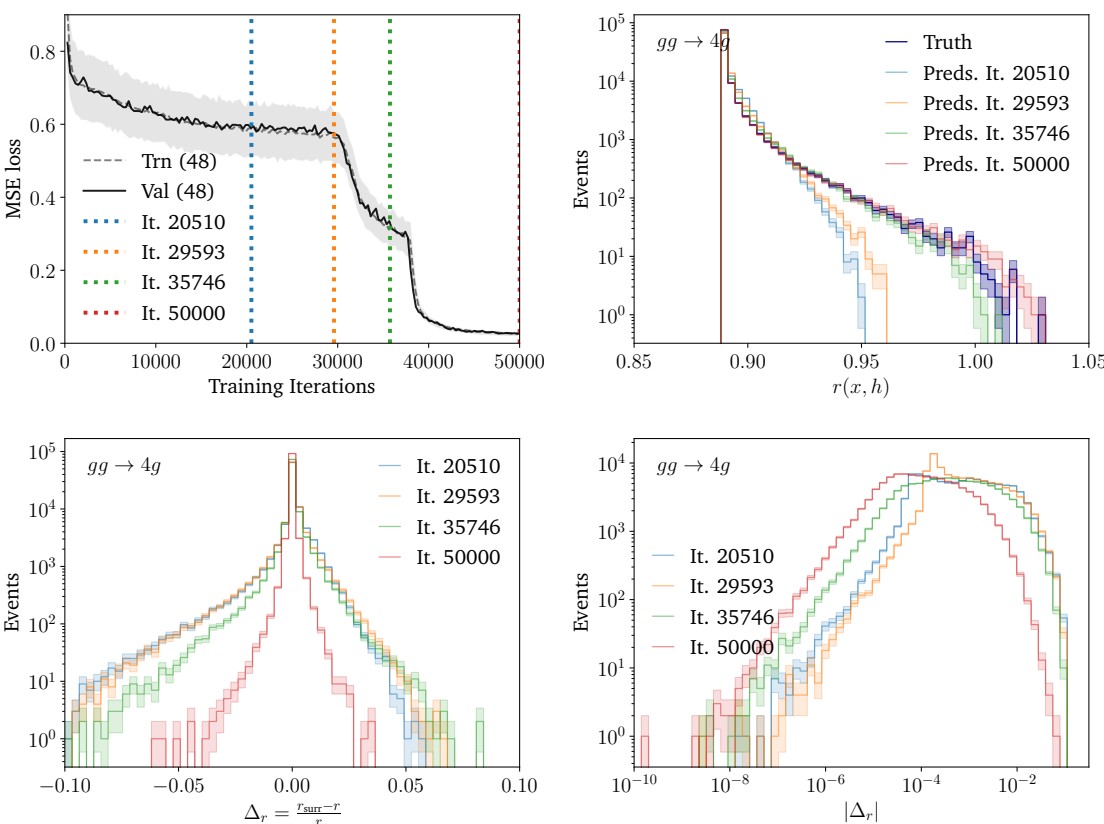

Figure 1: Loss collapse for the transformer trained on the $gg \to 4g$ dataset. We first show the loss curve and the collapses after 30k and 37k iterations. The vertical dashed lines indicate the checkpoints where we evaluate the network. Next, we show the target reweighting factor along with the predictions at different checkpoints, the relative deviation of the predictions, and its absolute value.

Throughout our study, we consistently observe a characteristic training pattern for the standard transformer: a long plateau with modest loss decrease, then a sudden drop in validation loss after a sizeable number of epochs. This drop appears in two stages. It does not appear for the MLP, the GNN, or L-GATr. This leads to the question whether the transformer first explores the parameter space and then defines an internal representation respecting the symmetry of the data.

This drop in loss can be controlled. The embedding dimension plays a role in how early and how reliably the loss collapse takes place within a given number of iterations. Enlarging the training batch size, improving the data preprocessing, or disabling dropout, all reduce the training time until the loss collapse. Reducing the magnitude of network initialization weights during initialization also triggers an early loss collapse. This behavior is similar to grokking [90, 91], where a network first memorizes the training data and then adjusts its latent representation to generalize.

Throughout this section, all networks are built with an embedding dimension of 48 and trained for 50k iterations. This is sufficient to go through the collapse phase. The task is to regress the standardized reweighting factor

$$\tilde{r}(x) = \frac{r - \bar{r}}{\sigma(r)} \qquad \text{for} \qquad gg \to 4g \ . \tag{26}$$

In Fig. 1, we show an example run where the loss collapses first after 30k training iterations.

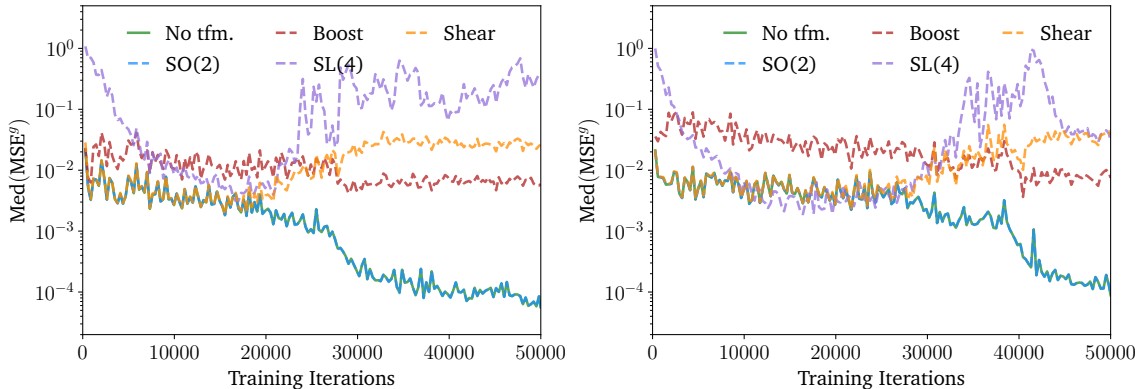

Figure 2: Losses Med($MSE^0$) (solid, green line) and Med($MSE^g$) (dashed lines) during training, for two example runs and our set of symmetry transformations $g$.

On top right and bottom left, we clearly see that up to approximately 30k iterations the network achieves modest accuracy and struggles with phase space regions with the largest targets. After 5k additional iterations, the network has undergone a drop in loss, enabling precise predictions over nearly the entire target domain. After 50k iterations and a second loss collapse the relative accuracy increases to the per-cent level.

To relate the loss collapse to learned symmetries, we need to see how the predictions change under input transformations. Our regression target is Lorentz-invariant, so we can check invariance under $g$ as a proxy for the special orthochronous Lorentz group $\Lambda \in SO^+(1,3)$. We also want to see how the network predictions change under symmetry transformations $g$ not related to the data. To this end, at each validation instance during training we evaluate the network for input transformed under different transformations $g$,

$$\tilde{r}_{\text{surr}}(x_g, h) \equiv \text{Transformer}(g(x), h) \,, \tag{27}$$

and compare the output to the original $\tilde{r}_{\text{surr}}(x, h)$.

Our set of Lorentz-like transformations include a random SO(2) rotation around the $z$-axis

$$g_{SO(2)} = \begin{pmatrix} 1 & 0 & 0 \\ 0 & R(\phi) & 0 \\ 0 & 0 & 1 \end{pmatrix} \qquad R(\phi) \in SO(2) \,, \tag{28}$$

and a general random Lorentz boost of the input four-momenta

$$\tilde{r}_{\text{surr}}(x_{\text{Boost}}, h) = \text{Transformer}(x_{\text{Boost}}, h), \qquad x_{\text{Boost}}^\mu = \Lambda^\mu_\nu x^\nu \,. \tag{29}$$

As transformations unrelated to the Lorentz group, we include a random rotation with unit determinant, $g \in SL(4)$, and a random shear where we mix $p_x$ with $p_y$ and $p_z$, and $p_y$ with $p_z$

$$g_{\text{shear}} = \begin{pmatrix} 1 & 0 & 0 & 0 \\ 0 & 1 & a & b \\ 0 & 0 & 1 & c \\ 0 & 0 & 0 & 1 \end{pmatrix} \qquad \text{with} \quad a, b, c \sim \mathcal{U}[-1, 1) \,. \tag{30}$$

We define the MSE losses without and with the transformation $g$ of the input as

$$\text{MSE}^0(\tilde{r}) = \left[ \tilde{r}_{\text{surr}}(x, h) - \tilde{r}(x, h) \right]^2$$
$$\text{MSE}^g(\tilde{r}) = \left[ \tilde{r}_{\text{surr}}(x_g, h) - \tilde{r}(x, h) \right]^2 \,. \tag{31}$$

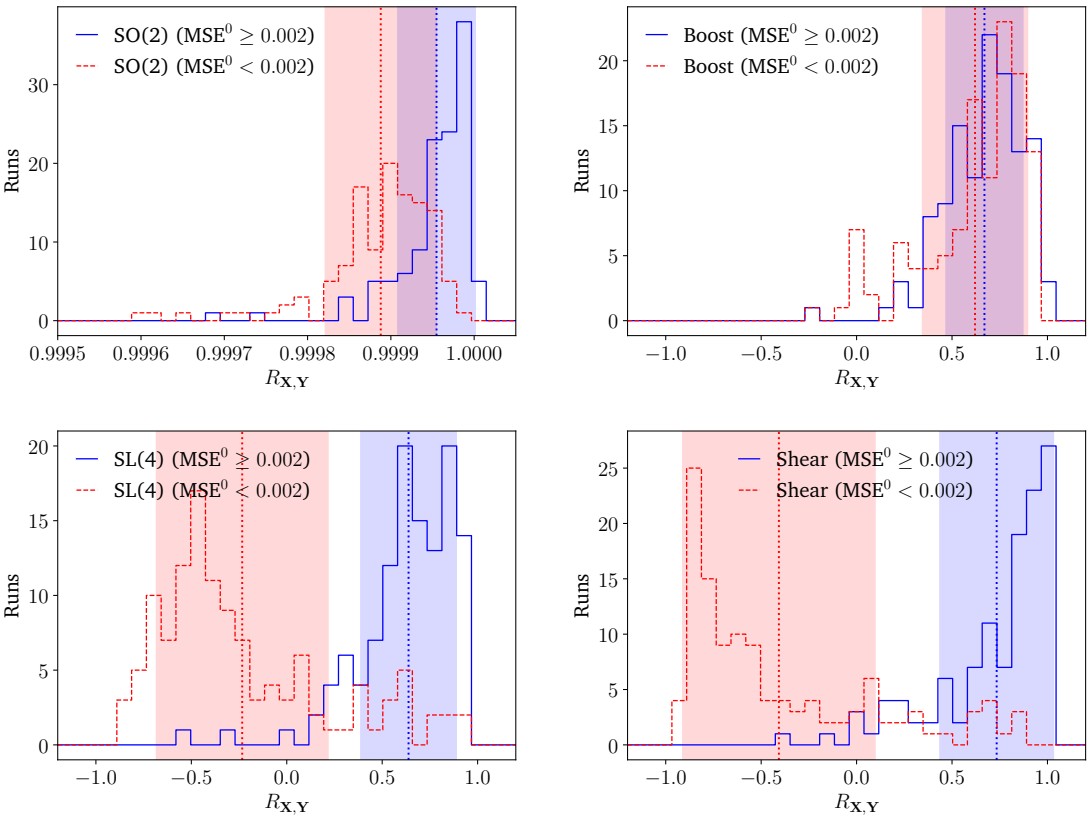

Figure 3: Pearson correlation coefficients between $\text{Med}(\text{MSE}^0)$ and $\text{Med}(\text{MSE}^g)$ for 120 identical runs for four symmetry transformations $g$. The blue, solid histograms and red, dashed ones represent the correlation coefficient before and after the loss collapse, respectively. Vertical dotted lines and shaded bands depict the mean and standard deviation of each histogram.

In Fig. 2 we show the median losses for each validation instance during training for two example runs. The medians are chosen for robustness against outliers. First, we see that the median loss under SO(2) rotations overlaps with the untransformed input loss. This is because it is the only transformation that leaves the initial-state transverse momenta $p_x$ and $p_y$ — which are always zero — unchanged.

Before the drop in the loss, the loss under shear transformations coincides with that of the original inputs, similar to SO(2). After the first transition the shear loss becomes anti-correlated with the baseline, a behavior also seen for SL(4). The second stage collapse around iteration 27k (left) and 37k (right) in Fig. 2 comes with a considerable decrease in the loss for the boosted input. Interestingly, prior to the loss collapse, the networks progressively learn the SL(4) invariance, while after the first loss drop, they have *learned* to penalize these rotations, as they are not Lorentz invariant.

Our results suggest that the transformer progressively aligns its internal representation to respect the symmetries in the data and improve its performance. For a final test, we go beyond isolated runs and perform 120 independent trainings with different seeds. For each network, we compute the Pearson correlation coefficient between the medians of the MSE losses of the original and the transformed input,

$$R_{\mathbf{X},\mathbf{Y}}(g) = \frac{\text{cov}(\mathbf{X}, \mathbf{Y})}{\sqrt{\text{var}(\mathbf{X})\text{var}(\mathbf{Y})}} \qquad \mathbf{X} = \text{Med}(\text{MSE}^0) \qquad \mathbf{Y} = \text{Med}(\text{MSE}^g) \,. \tag{32}$$

We compute two correlation coefficients for each transformation, one over the iterations preceding the first loss collapse, and one over the iterations following it. Since the collapse occurs at different iterations, we define the turning point as the first iteration where the untransformed input loss drops below $\text{Med}(\text{MSE}^0) = 0.002$. In Fig. 3 we show these correlation coefficients for our four symmetry transformations. The pre-collapse values are shown as solid blue histograms, while the post-collapse values appear as dashed red histograms.

This study clarifies that, statistically, the SO(2)-rotated and boosted input losses remain roughly equally correlated to the untransformed input loss throughout training. Since the network effectively minimizes $\text{MSE}^0$, it also minimizes $\text{MSE}^{\text{SO}(2)}$ and $\text{MSE}^{\text{Boost}}$. The SL(4) and shear losses are initially correlated with the untransformed input loss and turn into anti-correlated after the collapse. This confirms that the transformer learns a latent representation that progressively aligns with the symmetries in the data.

# 4 Unweighting performance

## 4.1 All-gluon processes

As a starting point, we look at the regressing of the reweighting factor. As a figure of merit, we consider the relative accuracy of the network given by

$$\Delta_r = \frac{r_{\text{surr}} - r}{r} \, , \tag{33}$$

shown in the upper left panel of Fig. 4. The simple MLP is the least precise of the networks, manifested in a larger spread and a small bias towards negative values. For all other networks, $\Delta_r$ is symmetric around zero. For the third unweighting step against FC, we turn to the relevant weight distribution $t = r/r_{\text{surr}}$ in the upper right panel of Fig. 4. Its spread determines the size of $\epsilon_t$ and critically affects the effective gain factor, as shown in Eq.(18). Again, the MLP has the broadest distribution with a longer tail for large weights, resulting in small unweighting efficiencies $\epsilon_t$. In contrast, the rest of the networks have narrower weight distributions and the highest efficiencies.

Looking at the regression targets in Fig. 4 (bottom left), we find a region in the far tail that is undershot by the MLP, explaining the asymmetric $\Delta_r$ distribution. All other networks, GNN (Fig. 4, bottom left), transformer, and L-GATr (Fig. 4 bottom right), show comparable and improved accuracy.

In Fig. 5 we show the corresponding results for the high-multiplicity $gg \rightarrow 7g$ process. Here, transformer, GNN, and L-GATr scale better with the number of particles than the MLP, visibly outperforming it. Moreover, the GNN and L-GATr accurately reproduce the sharp left tail visible in the bottom row of Fig. 5.

The gain factors for the $gg \rightarrow ng$ processes are presented in Fig. 6 (right). The quoted uncertainty on the gain factor reflects only timing variations. Specifically, we account for fluctuations in the CPU evaluation time of both LC and FC amplitudes, as well as the surrogates. For the amplitudes, we observe timing variations of up to 30% depending on the hardware. For the surrogates, the uncertainty is estimated as the standard deviation over five runs. L-GATr is evaluated only once, as its computationally expensive.

To visualize the scaling with the multiplicity in Fig. 6 we show the $t = r/r_{\text{surr}}$ weight distribution for L-GATr and increasing multiplicities. In the right panel of Fig. 6, we see that the effective gain factor for all networks increases with higher multiplicity. This is expected. The more expensive the evaluation of the FC amplitude becomes, the more we can save by

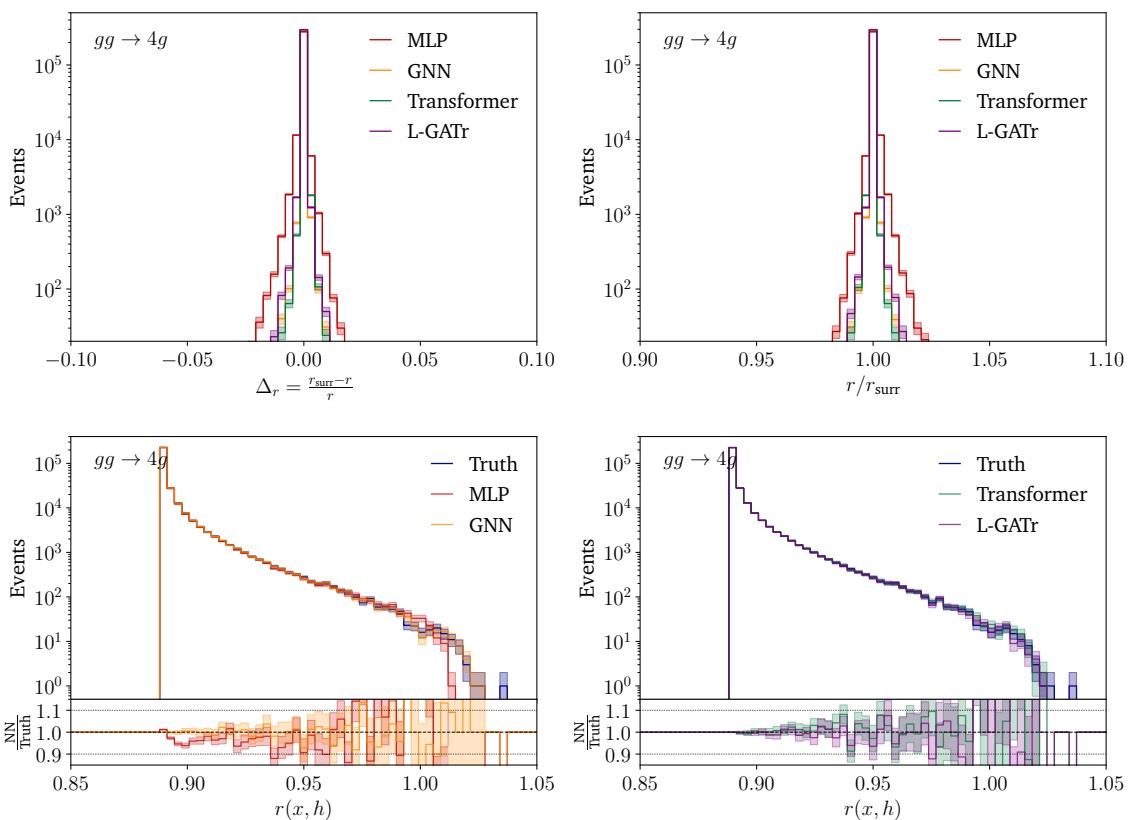

Figure 4: Top: relative accuracy $\Delta_r$ (left) and $t = r/r_{\mathrm{surr}}$ (right), used in the calculation of the second surrogate efficiency (right) for all networks and the $gg \to 4g$ process. Bottom: regression targets and predictions of the MLP and GNN (left) and transformer and L-GATr (right) for $gg \to 4g$.

evaluating a surrogate instead.

The MLP has an effective gain factor comparable or even better than the other networks, even though its regression performance is worse. This happens because the effective gain factor is not only a function of the unweighting efficiency, but also depends on the evaluation time of the surrogate. As the fastest network the MLP can compensate for its low accuracy. Altogether, we find speed-ups from about 15% for low multiplicities to more than a factor of two for the highest multiplicities. This sizeable improvement over the expensive FC matrix element computations will become important for precision studies at the HL-LHC. We expect even higher gain factors from LLoCa architectures [53] which are Lorentz-equivariant and achieve the same accuracy as L-GATr while being faster to train and to evaluate.

For a better intuition on the behavior of the effective gain factor, we include Fig. 10 in the Appendix. It illustrates the effect of the LC/FC amplitude evaluation time on the effective gain factor, given fixed unweighting efficiencies.

## 4.2 Single and double quark-line processes

In Fig. 7, we show the weight distribution $t = r/r_{\mathrm{surr}}$ for varying multiplicities using L-GATr (left), and the effective gain factors for all networks (right). The top panel shows the results for the single quark-line processes, observing and average speed-up of 30% and up to 50% for the highest multiplicity. For the processes involving two quark lines, we first show

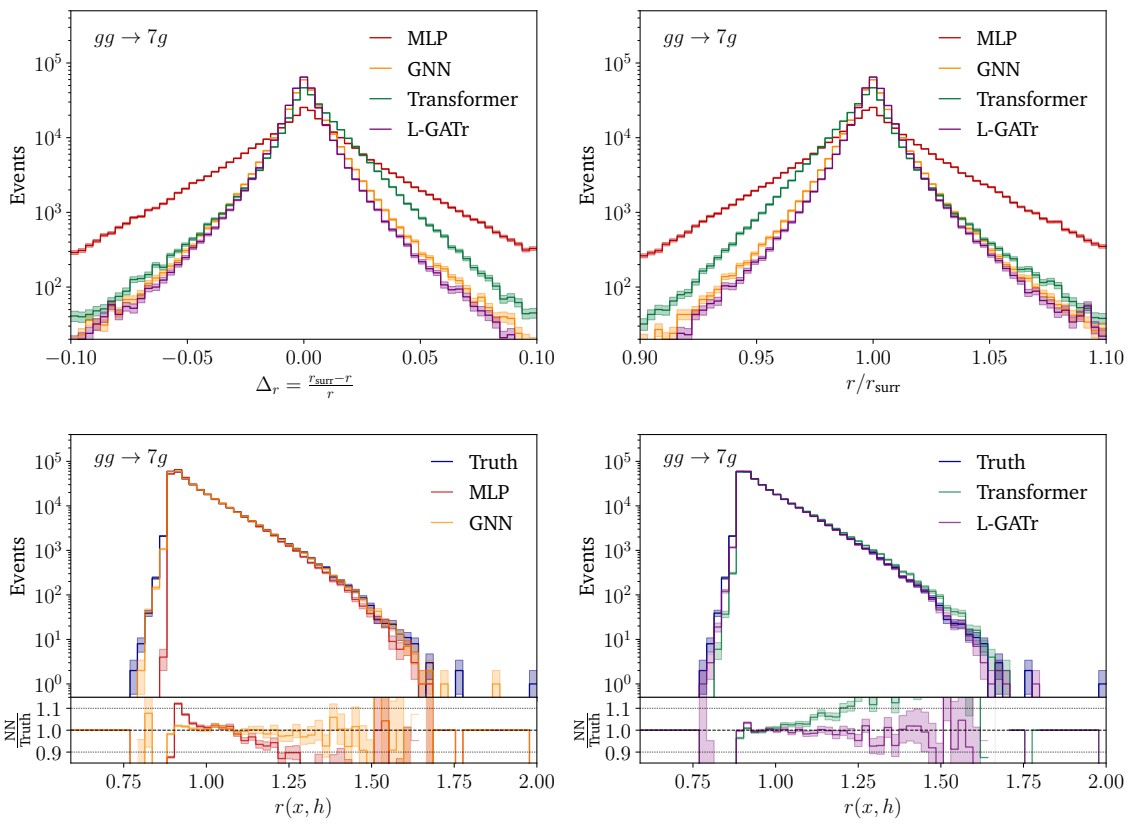

Figure 5: Top: relative accuracy $\Delta_r$ (left) and $t = r/r_{\text{surr}}$ (right), used in the calculation of the second surrogate efficiency (right) for all networks and the $gg \rightarrow 7g$ process. Bottom: regression targets and predictions of the MLP and GNN (left) and transformer and L-GATr (right) for $gg \rightarrow 7g$.

$d\bar{d} \rightarrow u\bar{u} + (n-2)g$ in the center panels, and $gg \rightarrow d\bar{d}u\bar{u} + (n-4)g$ in the bottom panels, each for two representative color orderings for the different kinematics of the total cross section. For the $d\bar{d} \rightarrow u\bar{u} + (n-2)g$ case, we obtain an average speed-up of about 80% across multiplicities and color orderings, reaching 80-100% effective gains in the most expensive multiplicity. Finally, for the $gg \rightarrow d\bar{d}u\bar{u} + (n-4)g$ process in the bottom row, we reach about 75-90% effec-

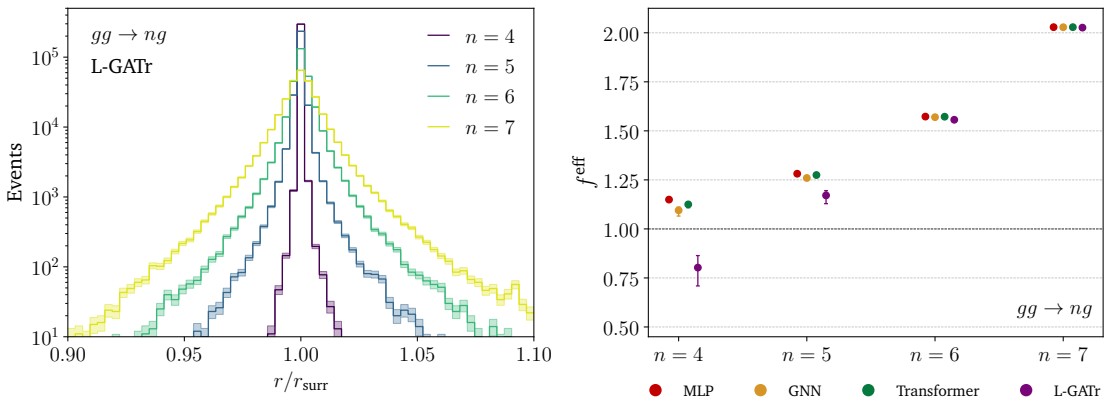

Figure 6: Gain factors for our surrogate approach described in Alg. 2, for processes involving only gluons for varying multiplicity.

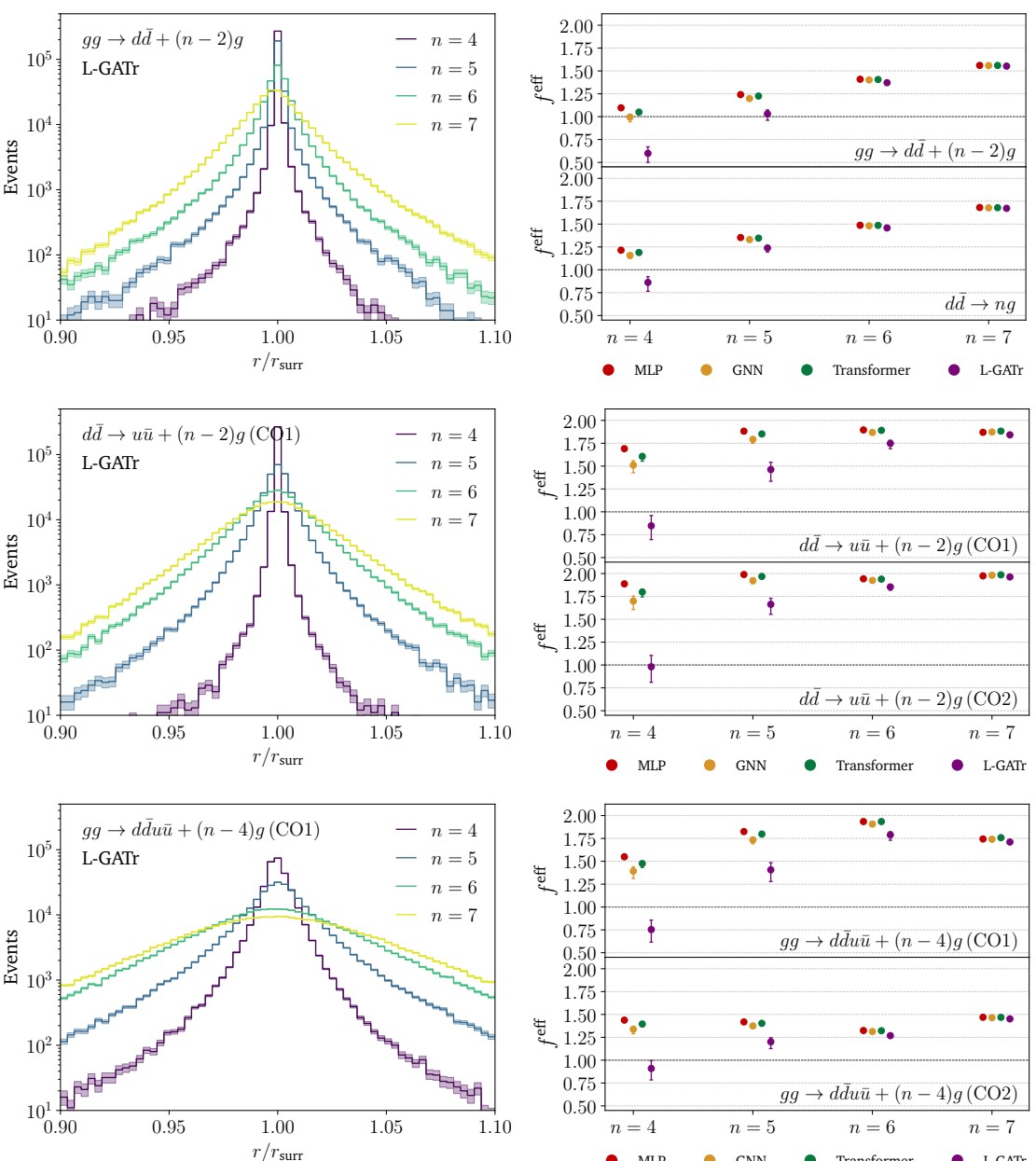

Figure 7: Spread of $t = r/r_{\text{surr}}$ for L-GATr (left), and effective gain factors from our surrogate method described in Alg. 2 for all tested networks (right). Results are shown for the single quark-line processes: $gg \to d\bar{d} + (n-2)g$ and $d\bar{d} \to ng$ (top), and the two double quark-line processes: $d\bar{d} \to u\bar{u} + (n-2)g$ (middle) and $gg \to d\bar{d}u\bar{u} + (n-4)g$ (bottom).

tive gain factors for CO1. For CO2, on the other hand, they remain around 50%. Nonetheless, the average effective gain for this specific color channel is found to be roughly 35%, which is still substantial. Overall, averaging over all networks, processes and multiplicities, we find an effective gain factor of about 50%, effectively demonstrating the that the use of surrogates for QCD event generation at FC accuracy is possible and resource-saving.

As can be seen from Fig. 16 in the Appendix, the reweighting factors for this specific ordering feature a narrow peak with a short right tail and a long left tail. This indicates that the FC unweighting efficiency $\epsilon_r$ is relatively high, so the effective gains are limited by an already

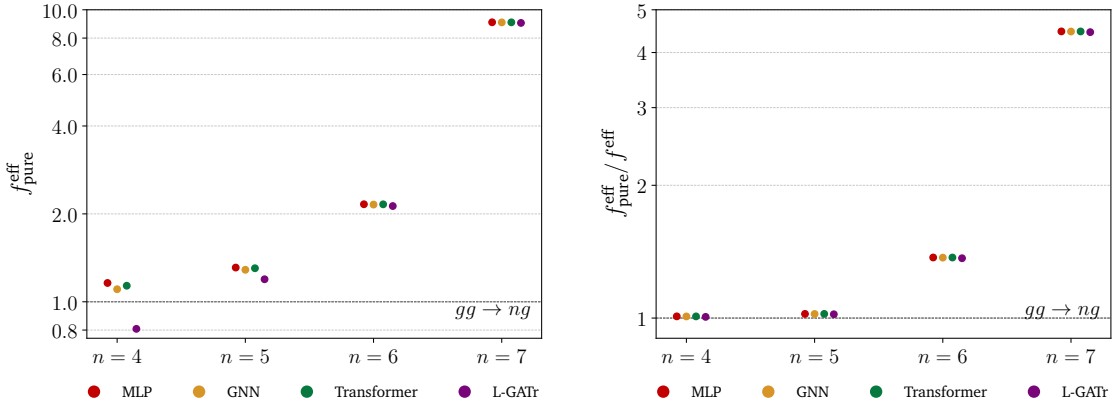

Figure 8: Pure surrogate unweighting effective gain factor (left) and relative gain compared to the standard three-step procedure (right), for all networks and varying multiplicities of the $gg \rightarrow ng$ process.

efficient LC-based unweighting. Supporting this interpretation, we note that $d\bar{d} \rightarrow 4g$ and, in particular, $gg \rightarrow d\bar{d} + 2g$ exhibit similar distributions and also yield reduced gains.

## 4.3 Pure surrogate unweighting

In our current three-step setup, we still evaluate the FC amplitude as part of the final step. This can be omitted and events can be unweighted against the surrogate, provided the surrogate accuracy and uncertainty are under control and their impact is below other systematics, such as neglected higher orders or large logarithms. We note that this does not only require extremely accurate regression networks, but also a calibrated uncertainty with no significant tails in the pull distribution.

Under the assumption that the final FC reweighting is not needed, the effective gain factor simplifies to

$$f_{\text{pure}}^{\text{eff}} = \alpha \, \frac{\dfrac{1}{\epsilon_{\text{LC}}\epsilon_r}\langle t_{\text{LC}}\rangle + \dfrac{1}{\epsilon_r}\langle t_{\text{FC}}\rangle}{\dfrac{1}{\epsilon_{\text{LC}}\epsilon_{\text{surr}}}\langle t_{\text{LC}}\rangle + \dfrac{1}{\epsilon_{\text{surr}}}\langle t_{\text{surr}}\rangle} \, . \tag{34}$$

In Fig. 8, we illustrate how moving to such a two-step algorithm would affect the effective gain factor. In the left panel of Fig. 8 we see that for high multiplicities, where the FC amplitude becomes expensive, the gain factor grows for the $gg \rightarrow 7g$ process.

To resolve the additional gain from unweighting only against a controlled ML surrogate, we show the relative improvement compared to the default three-stage unweighting in the right panel of Fig. 8. For low multiplicities, where evaluating the LC and FC amplitudes is not that expensive, there is no visible improvement over the three-step algorithm. For high multiplicities, we gain increasingly more from not evaluating the FC amplitude. This showcases a promising future direction and defines a challenge for accurate and precise ML-surrogates over all of phase space.

# 5 Outlook

This study demonstrates that ML surrogates are a powerful and versatile tool to accelerate LHC event generation at FC accuracy. Building on the existing two-step LC-to-FC approach, where events are first generated in the leading-color approximation and subsequently reweighted, we insert an additional surrogate step into the unweighting chain. It yields effective speed-ups of up to a factor of two for representative high-multiplicity processes, while fully retaining FC accuracy.

Our comparison of network architectures shows that symmetry-aware designs such as Lorentz-equivariant transformers are particularly well suited, with robust scaling toward larger multiplicities. At the same time, the MLP is less accurate in reproducing the reweighting factor but achieves comparable gain factors due to its speed. This interplay between accuracy and evaluation time highlights the potential of future architectures such as LLoCa, which combine the accuracy of Lorentz-equivariant designs with reduced computational cost.

A promising future direction of study is pure surrogate unweighting, where the final FC reweighting step can be skipped. This requires stringent control over accuracy and uncertainties, with potential gain factors close to an order of magnitude for high multiplicities. This shortcut should become feasible once calibrated surrogate uncertainties are fully integrated into the event generation pipeline, providing reliable uncertainty estimates propagated into event predictions.

# Acknowledgments

We thank Anja Butter and Olivier Mattelaer for their contribution during an earlier phase of the project, and Víctor Bresó-Pla for many insights and help with the L-GATr implementation. We thank Jonas Spinner for valuable suggestions on the GNN implementation, advice regarding L-GATr, and detailed comments on the manuscript. We would also like to thank Sofia Palacios Schweitzer and Ayodele Ore for their very helpful comments. J.M.V. is funded by the BMBF Junior Group *Generative Precision Networks for Particle Physics* (DLR 01IS22079). The Heidelberg group is supported by the Deutsche Forschungsgemeinschaft (DFG, German Research Foundation) under grant 396021762 – TRR 257 *Particle Physics Phenomenology after the Higgs Discovery*. This work was also supported by the DFG under Germany's Excellence Strategy EXC 2181/1 - 390900948 *The Heidelberg STRUCTURES Excellence Cluster*. The authors acknowledge support by the state of Baden-Württemberg through bwHPC and the German Research Foundation (DFG) through grant no INST 39/963-1 FUGG (bwForCluster NEMO). R.F. and T.V. are supported by the Swedish Research Council under contract numbers VR:2020-04423 and VR:2023-00221, respectively. Parts of the computations were enabled by resources within the project UPPMAX 2025/2-312 provided by the National Academic Infrastructure for Supercomputing in Sweden (NAISS).

# A Additional plots

We show additional regression results for all considered processes introduced before. In each figure, we show on the left column the distribution of targets and network predictions, while on the right column, we display the ratio $r/r_{surr}$ distributions, whose variance influences directly the secondary surrogate unweighting efficiency $\epsilon_t$. The results are ordered as follows: the $gg \to ng$ dataset corresponds to the Fig. 9, $gg \to d\bar{d} + (n-2)g$ to Fig. 11, $d\bar{d} \to ng$ to Fig. 12, $d\bar{d} \to u\bar{u} + (n-2)g$ to Figs. 13, and 14 for two representative color orderings, and $gg \to d\bar{d}u\bar{u} + (n-4)g$ to Figs. 15 and 16, also for two representative color orderings.

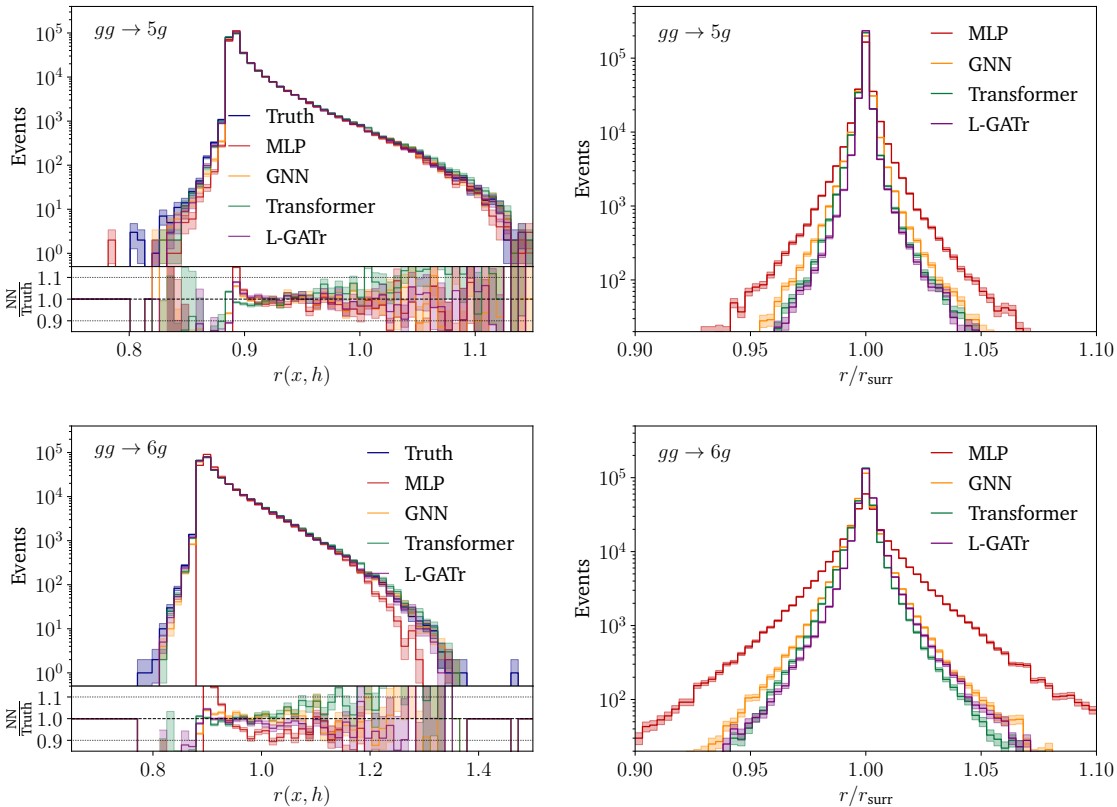

Figure 9: Regression results for $gg \to ng$ for the four networks considered and $n \in \{5, 6\}$. We show, on the left column, the distributions of true and network predictions, and the ratio $r/r_{surr}$ distribution on the right column. We skip here the $n = 4$ and $n = 7$ processes, already shown in the main text in Figs. 4, and 5, respectively.

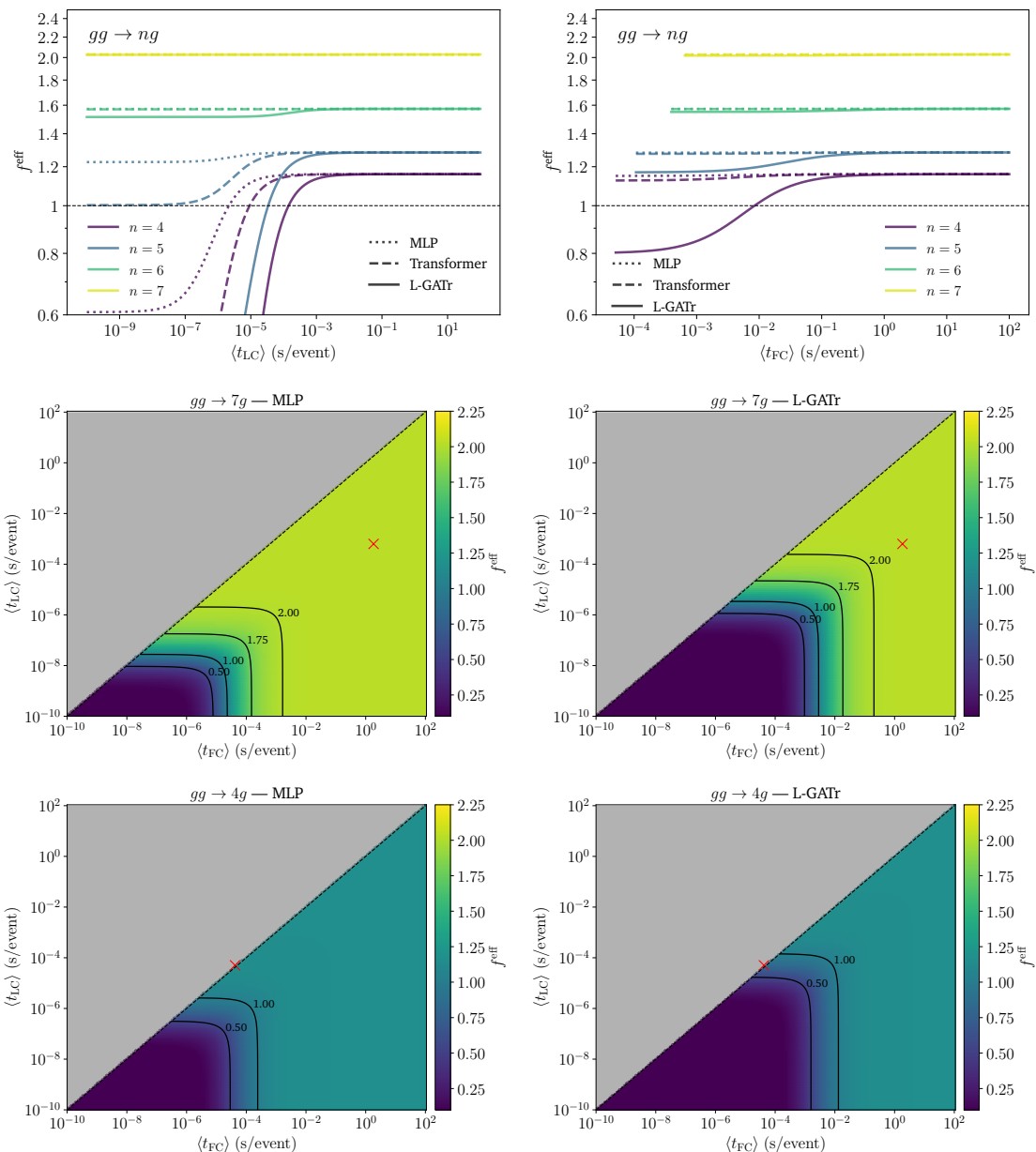

Figure 10: Dependence of the effective gain factors on the amplitudes evaluation time at LC and FC precision. On the top left (right), we show, for three networks and the four multiplicities of the $gg \to ng$ processes, the effective gains as a function of LC (FC) amplitude evaluation time. For the left (right) plot, we fixed all the values to the measurements obtained in this work, except for $\langle t_{LC} \rangle$ ($\langle t_{FC} \rangle$). On the right, we also constrain $\langle t_{FC} \rangle > \langle t_{LC} \rangle$, since the opposite is not physical. On the middle (bottom) row we show the joint dependence on both evaluation times for the $gg \to 7g$ ($gg \to 4g$) process, for the MLP and L-GATr in separate columns. We show as red crosses the measurements obtained in this work.

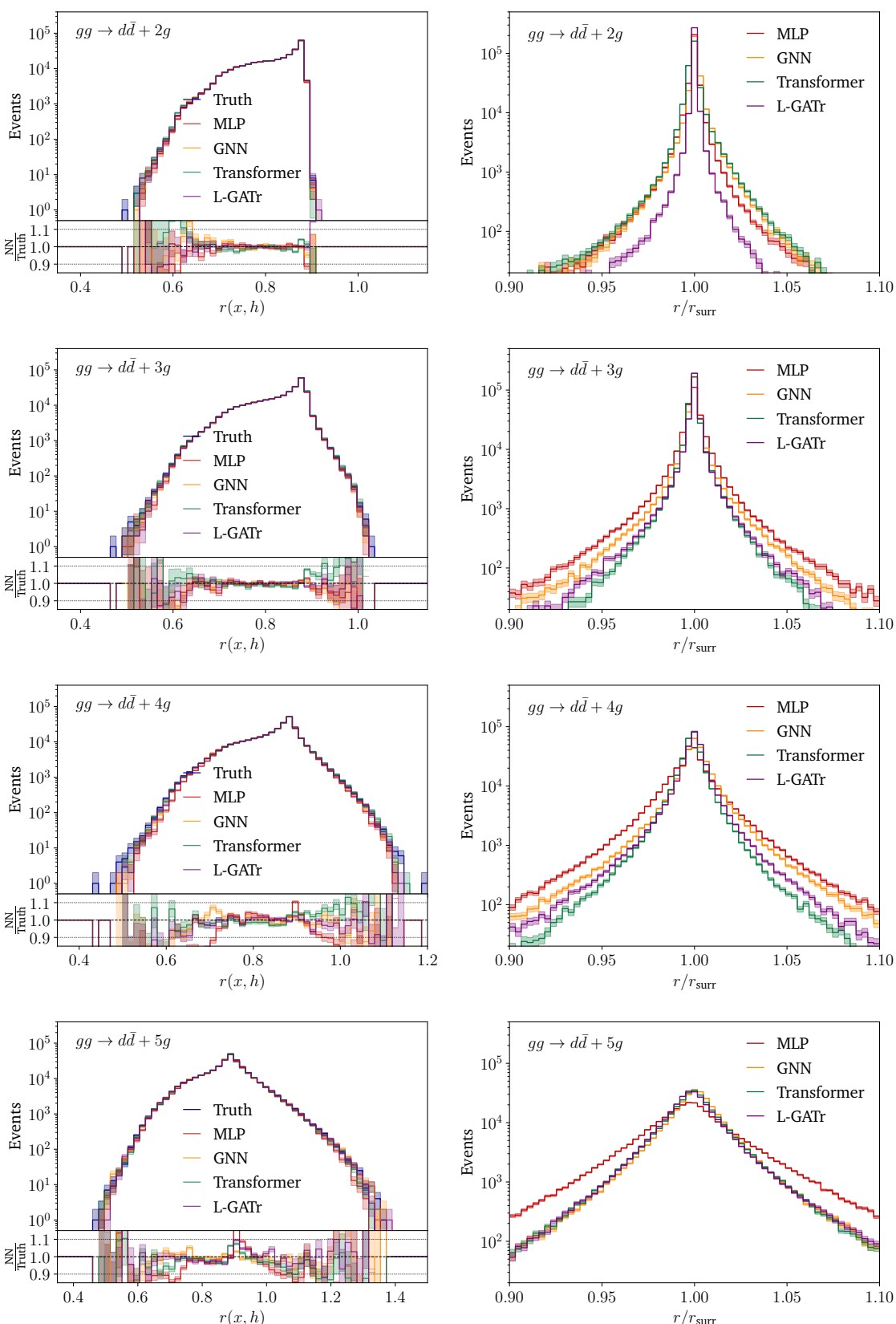

Figure 11: Regression results for $gg \rightarrow d\bar{d} + (n-2)g$ for the four networks considered and $n \in \{4, 5, 6, 7\}$. We show, on the left column, the distributions of true and network predictions, and the ratio $r/r_{\text{surr}}$ distribution on the right column.

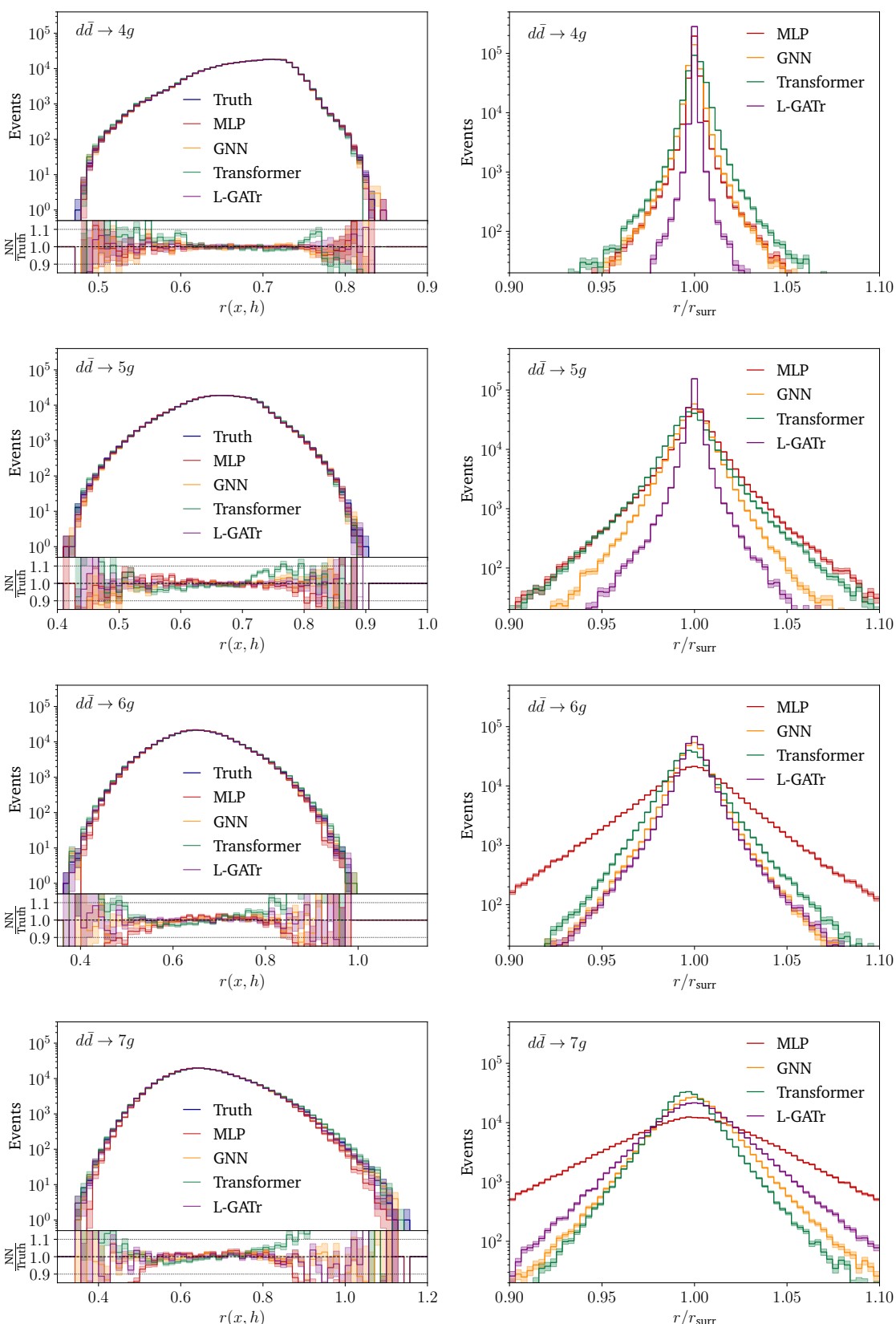

Figure 12: Regression results for $d\bar{d} \rightarrow ng$ for the four networks considered and $n \in \{4, 5, 6, 7\}$. We show, on the left column, the distributions of true and network predictions, and the ratio $r/r_{\text{surr}}$ distribution on the right column.

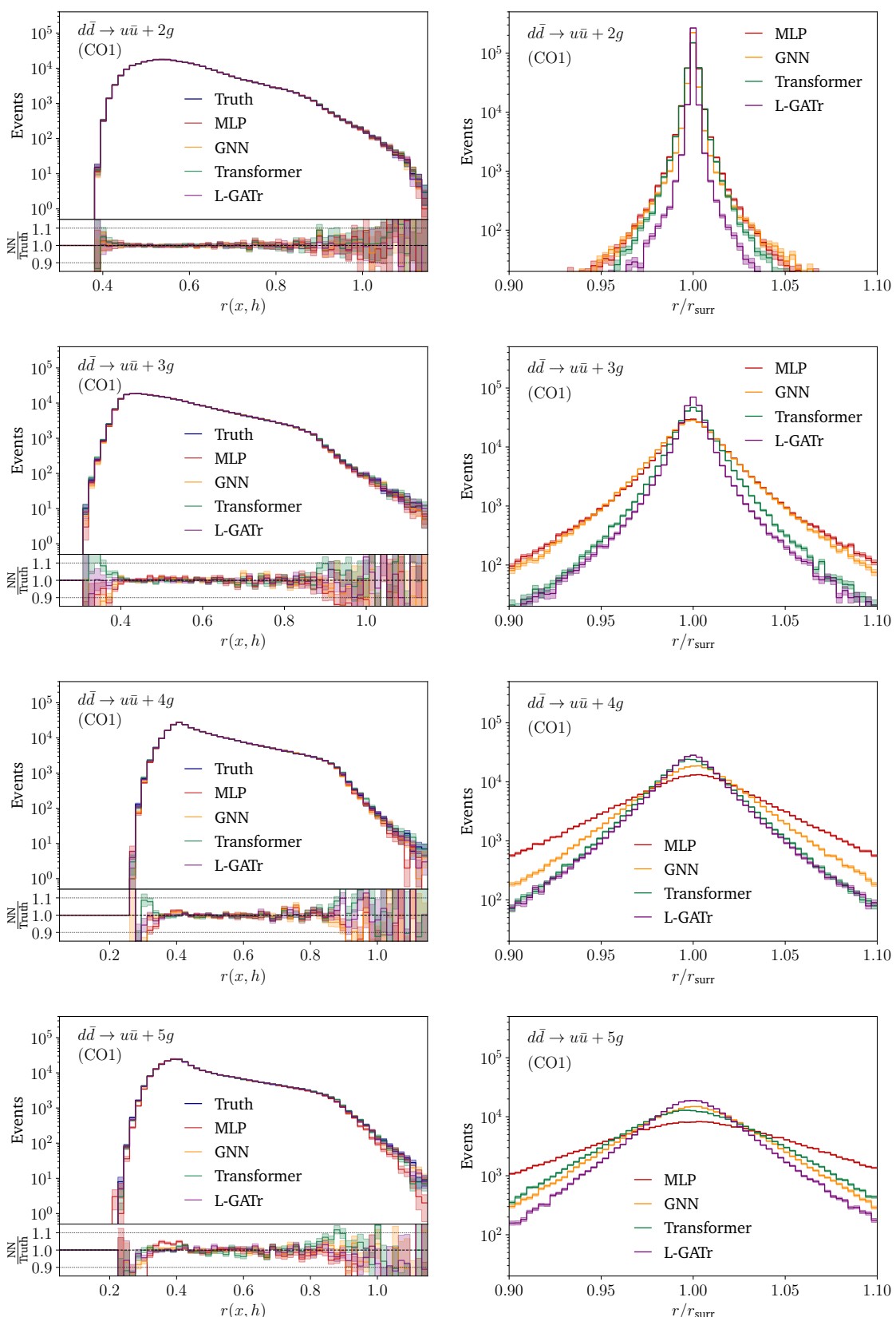

Figure 13: Regression results for $d\bar{d} \to u\bar{u} + (n-2)g$ (CO1) for the four networks considered and $n \in \{4, 5, 6, 7\}$. We show, on the left column, the distributions of true and network predictions, and the ratio $r/r_{\text{surr}}$ distribution on the right column.

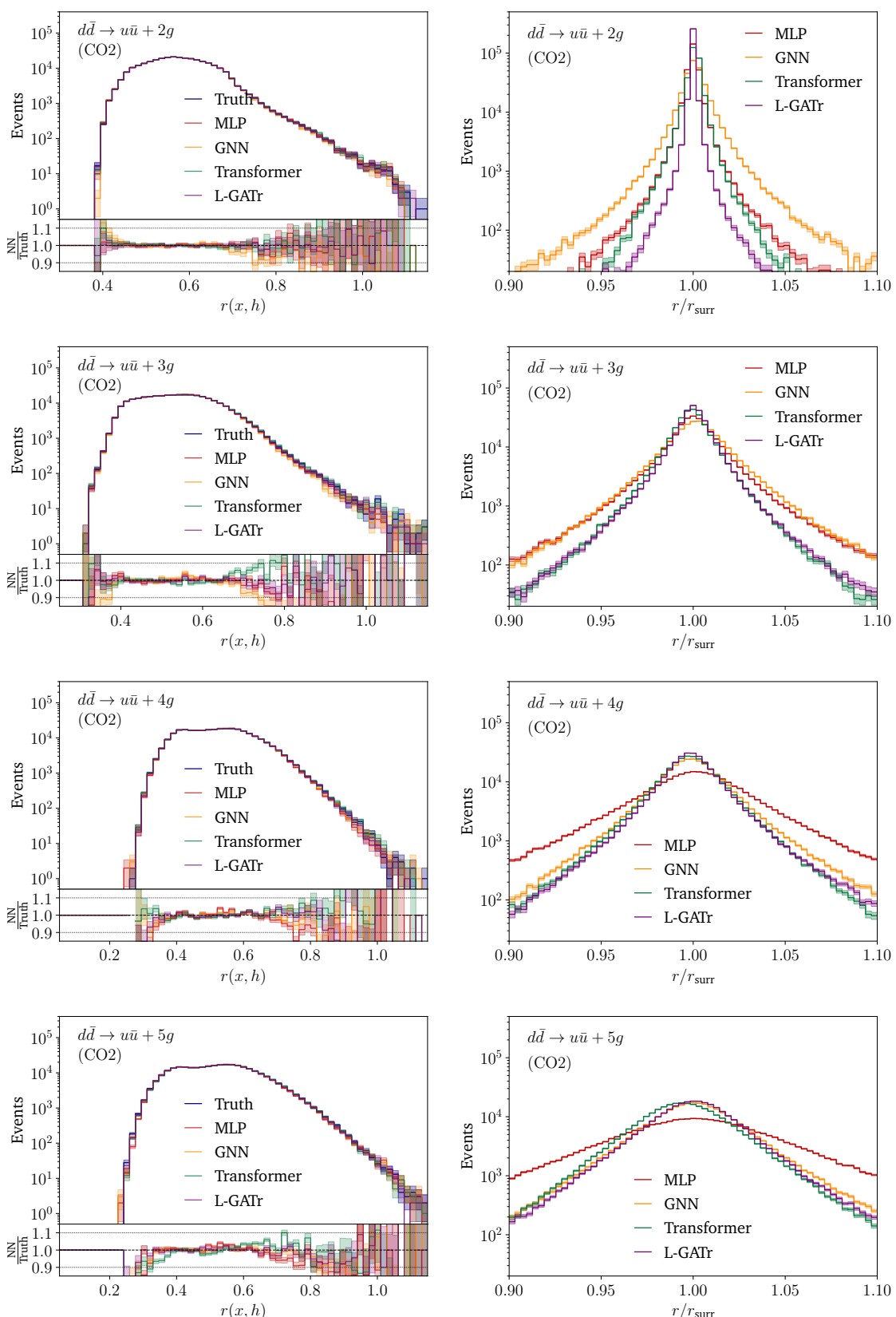

Figure 14: Regression results for $d\bar{d} \to u\bar{u} + (n-2)g$ (CO2) for the four networks considered and $n \in \{4, 5, 6, 7\}$. We show, on the left column, the distributions of true and network predictions, and the ratio $r/r_{\text{surr}}$ distribution on the right column.

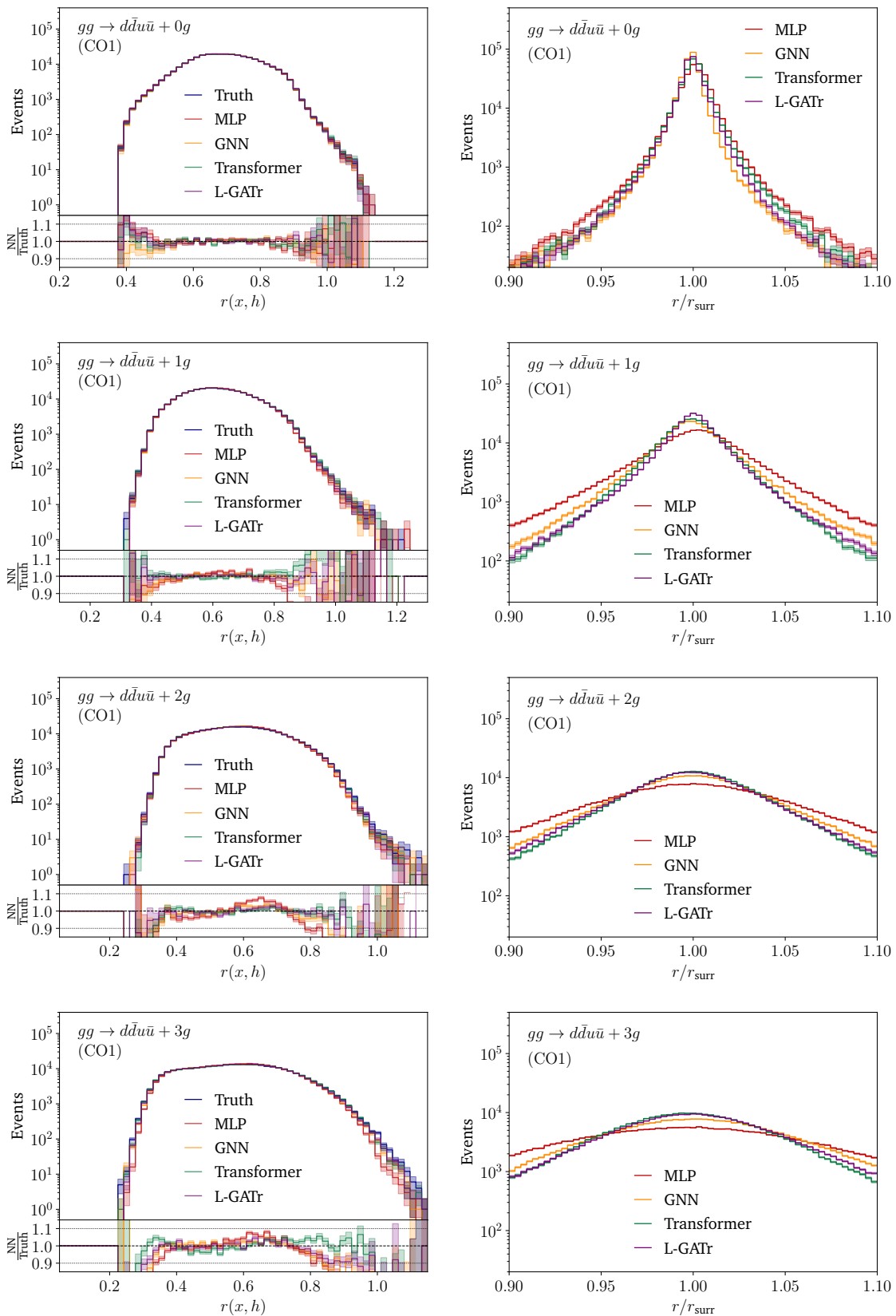

Figure 15: Regression results for $gg \to d\bar{d}u\bar{u} + (n-4)g$ (CO1) for the four networks considered and $n \in \{4, 5, 6, 7\}$. We show, on the left column, the distributions of true and network predictions, and the ratio $r/r_{surr}$ distribution on the right column.

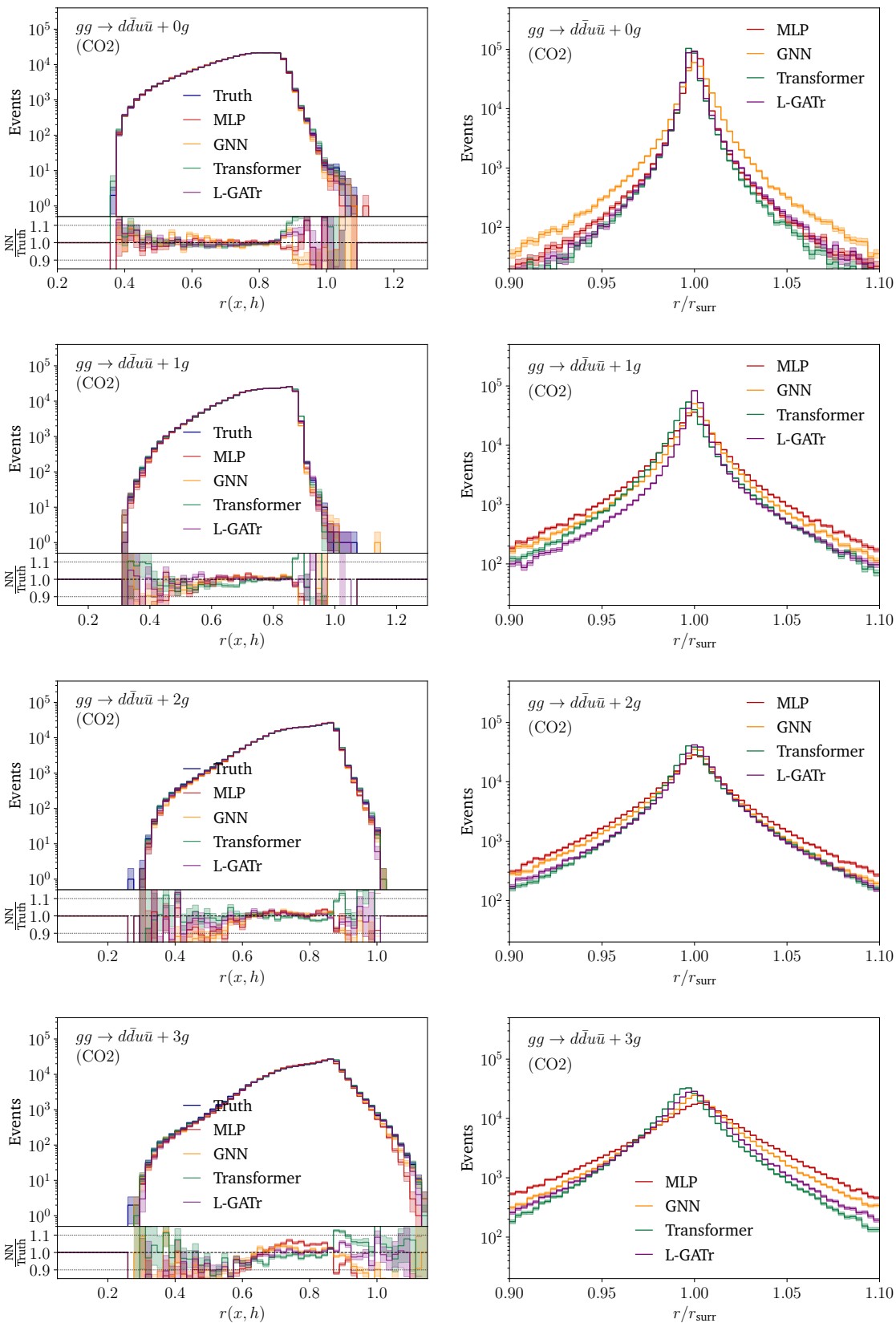

Figure 16: Regression results for $gg \to d\bar{d}u\bar{u} + (n-4)g$ (CO2) for the four networks considered and $n \in \{4, 5, 6, 7\}$. We show, on the left column, the distributions of true and network predictions, and the ratio $r/r_{surr}$ distribution on the right column.

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
