# Peer review of "FASTColor -- Full-color Amplitude Surrogate Toolkit for QCD"

_SciPost Physics_

## Round 1 · Referee Report · Anonymous (Referee 1) · 2025-12-11

Report

FASTColor – Full-color Amplitude Surrogate Toolkit for QCD

Both the generation of particle collision processes as well as the simulation of their interaction with detectors is both crucially important and resource intensive. This work investiages how leading-color (LC) simulations can be upscaled to full-color (FC) precision using a machine-learning based reweighting scheme.

The paper is very well written, and carries out numerical studies that demonstrate the gain (up to a factor of 4-5 in the best case) over previously established techniques.

Below are listed two minor points that might be addressed, but overall this paper would be a good publication for SciPost Physics.

-> Rigth now, the production of the training sample (for which both LC and FC are required) and the neural network training are treated as 'free'. Can the necessary compute be quantified and put into relation to the expected number of samples that will be 'upsampled'

-> Relatedly, a discussion how reuseable vs problem specific these reweighting networks are envisioned? Is this currently e.g. a separate network per process, or is there some synergy (realised or conceivable)

Recommendation

Ask for minor revision

---

## Editorial Decision

in_refereeing